# Similarity Preserving Transformer Cross-Modal Hashing for Video-Text Retrieval

## ABSTRACT

As social networks grow exponentially, there is an increasing demand for video retrieval using natural language. Cross-modal hashing that encodes multi-modal data using compact hash code has been widely used in large-scale image-text retrieval, primarily due to its computation and storage efficiency. When applied to video-text retrieval, existing unsupervised cross-modal hashing extracts the frame- or word-level features individually, and thus ignores long-term dependencies. In addition, effective exploit of multi-modal structure poses a significant challenge due to intricate nature of video and text. To address the above issues, we propose Similarity Preserving Transformer Cross-Modal Hashing (SPTCH), a new unsupervised deep cross-modal hashing method for video-text retrieval. SPTCH encodes video and text by bidirectional transformer encoder that exploits their long-term dependencies. SPTCH constructs a multi-modal collaborative graph to model correlations among multi-modal data, and applies semantic aggregation by employing Graph Convolutional Network (GCN) on such graph. SPTCH designs unsupervised multi-modal contrastive loss and neighborhood reconstruction loss to effectively exploit inter- and intra-modal similarity structure among videos and texts. The empirical results on three video benchmark datasets demonstrate that the proposed SPTCH generally outperforms state-of-the-arts in video-text retrieval.

## CCS CONCEPTS

• **Information systems → Multimedia and multimodal retrieval**.

## KEYWORDS

Hashing, Contrastive learning, Video-text Retrieval

## 1 INTRODUCTION

With the rapid development of social networks and short video sharing platforms, the number of videos on the web has exploded. When searching using natural language, it is desirable to retrieve relevant videos in a timely and accurate manner. However, multimedia data in different modalities exhibit significant structural differences, making it difficult to retrieve relevant content from a

Permission to make digital or hard copies of all or part of this work for personal or classroom use is granted without fee provided that copies are not made or distributed for profit or commercial advantage and that copies bear this notice and the full citation on the first page. Copyrights for components of this work owned by others than the author(s) must be honored. Abstracting with credit is permitted. To copy otherwise, or republish, to post on servers or to redistribute to lists, requires prior specific permission and/or a fee. Request permissions from permissions@acm.org.

*ACM MM, 2024, Melbourne, Australia*

© 2024 Copyright held by the owner/author(s). Publication rights licensed to ACM.
ACM ISBN 978-x-xxxx-xxxx-x/YY/MM
https://doi.org/10.1145/nnnnnnn.nnnnnnn

large amount of heterogeneous data. Therefore, efficient and effective cross-modal retrieval from large-scale multi-modal data has become a challenging problem.

Hashing [10, 12, 30] has been widely applied to large-scale cross-modal retrieval due to its efficiency in computation and storage. The idea of hashing is to project high-dimensional data into compact hash codes while preserving similarity among original data in the Hamming space. Cross-modal hashing maps multi-modal data into a common Hamming space to enable fast cross-modal retrieval, e.g., video-text retrieval. Supervised cross-modal hashing [4, 13, 36] requires high-quality semantic labels to supervise training, which are very expensive and time-consuming to obtain in real applications. Therefore, unsupervised cross-modal hashing [3, 16, 25, 38, 39] that does not rely on labels is extensively applied yet remains challenging.

Unsupervised deep cross-modal hashing [10, 31, 35] performs joint learning of feature and latent hash code in an end-to-end manner by optimizing an unsupervised loss. Conventional unsupervised deep cross-modal hashing methods are primarily tailored for image-text retrieval, and they encounter several main challenges when extended to video-text retrieval. Existing cross-modal hashing uses RNN [15] or LSTM [5] for video encoding. However, training LSTM is computationally expensive and also struggles to capture long-term dependencies among distant frames effectively due to gradient vanishing [19]. It is difficult to model similarity structure among multi-modal data, as label semantics are not available. In addition, it is not sufficient to effectively capture inter-modality and intra-modality similarity among video and text modalities due to their complex data structures. Therefore, it is challenging to develop unsupervised deep cross-modal hashing specifically designed for video-text retrieval.

To address the above concerns, we propose a new unsupervised deep cross-modal hashing method, i.e., Similarity Preserving Transformer Cross-Modal Hashing (SPTCH) for video-text retrieval. SPTCH encodes and reconstructs video and text using bidirectional transformer auto-encoder. SPTCH further exploits multi-modal structure by leveraging the superiority of contrastive learning. The overview of the proposed SPTCH is illustrated in Figure 1. The main contributions of this work are summarized:

- We propose Similarity Preserving Transformer Cross-Modal Hashing (SPTCH) for video-text retrieval. SPTCH utilizes the bidirectional transformer to effectively capture long-term dependencies in frame and word sequences. To our knowledge, SPTCH is among the first attempts of unsupervised deep cross-modal hashing specifically designed for video-text retrieval.
- SPTCH employs GCN on the constructed multi-modal collaborative graph to aggregate semantics of multi-modal data. SPTCH learns hash function and hash code by minimizing semantic reconstruction loss, neighborhood reconstruction loss, and multi-modal contrastive loss.

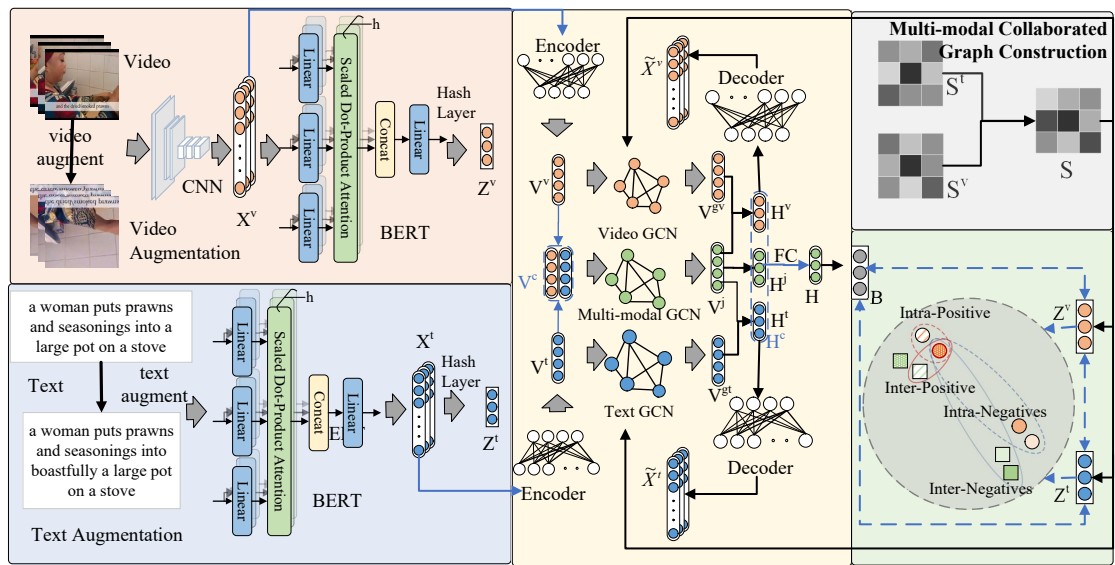

**Figure 1: Illustration of the proposed SPTCH. SPTCH employs the bidirectional transformer encoder to encode video and text, and then constructs a hash layer to transform continuous feature into hash code. SPTCH constructs a multi-modal collaborative graph to model correlations among multi-modal data, and applies semantic aggregation by employing Graph Convolutional Network (GCN) on such graph. SPTCH learns hash function and hash code by minimizing several losses, e.g., semantic reconstruction loss, neighborhood reconstruction loss, and multi-modal unsupervised contrastive loss.**

- The quantitative and qualitative results empirically verify the superiority of the proposed method over state-of-the-arts in video-text retrieval.

## 2 RELATED WORK

### 2.1 Cross-modal Hashing

Cross-modal hashing aims to map multi-modal data, mainly including images and texts, into a shared Hamming space, where hash codes of different modalities can be quickly compared and matched. The primary advantage of cross-modal hashing lies in its high efficiency in supporting cross-modal retrieval, mainly for image-text retrieval tasks. This work is closely related to unsupervised cross-modal hashing, where label semantics are unavailable in the training stage.

The shallow cross-modal hashing [3, 28] learns linear function to transform multi-modal data into hash code. For instance, Collective Matrix Factorization Hashing (CMFH) [3] jointly learns uniform hash codes and hash functions by using the cooperative matrix factorization technique. Semantic Topic Multimodal Hashing (STMH) [28] learns a common subspace by capturing multiple semantic topics from multimedia data and then generates binary hash codes by examining the representation of the semantic topics. The shallow cross-modal hashing typically uses hand-crafted features, and feature learning is independent of hash code learning. Therefore, the retrieval performance of shallow cross-modal hashing is expected to be further improved.

Unsupervised deep cross-modal hashing [11, 16, 25, 38, 39] jointly performs feature learning and hash code learning in an end-to-end

manner. Deep Binary ReConstruction (DBRC) [11] jointly performs heterogeneous modal correlation modeling and hash code learning in a binary reconstruction framework. Deep Joint-Semantics Reconstructing Hashing (DJSRH) [25] constructs a joint semantic relation matrix by integrating different similarity matrices and reconstructs such a joint matrix using hash code to preserve the semantic structure. Joint-modal Distribution-based Similarity Hashing (JDSH) [16] constructs a joint-modal similarity matrix to preserve cross-modal semantic correlation and generates hash code using a sampling and weighting scheme. Aggregation-based Graph Convolutional Hashing (AGCH) [38] generates a multi-modal similarity matrix by aggregating different similarity measures and employs GCN to preserve the semantic structure. Correlation-Identity Reconstruction Hashing (CIRH) [39] constructs a multi-modal collaborative graph to model heterogeneous multi-modal correlations and jointly performs intra-modal and inter-modal semantic aggregation on homomorphic and heteromorphic graph convolutional networks. Existing unsupervised cross-modal hashing primarily focuses on image-text retrieval, while video-text retrieval is becoming increasingly important due to the richer information of videos. Therefore, developing effective unsupervised deep cross-modal hashing for video-text retrieval remains challenging and deserves further research.

### 2.2 Video Hashing

Video hashing [8, 9, 14, 23, 32, 34, 37? ] aims to learn hash function and hash code that can yield impressive video retrieval performance.

Early attempts directly apply conventional image hashing methods, e.g., Spectral Hashing (SH) [30] and Anchor Graph Hashing (AGH) [17] on pooled video features, and treat video as a simple aggregation of independent frames and neglect the temporal information in videos. Later some shallow video hashing methods [34] are proposed to consider temporal information. To our knowledge, Video Hashing with both Discriminative commonality and Temporal consistency (VHDT) [34] is the first work that considers video structure. However, the capability of shallow hashing for video representation is limited.

Later deep video hashing [7–9, 14, 24, 32, 37 **?** ] has been proposed to extract improved video representation, due to the powerful capability of neural networks. Deep Video Hashing (DVH) [14] applies a Convolutional Neural Network (CNN) to extract features from each frame and leverages temporal and discriminative information within the video. It is of great significance to develop new deep hashing models that can preserve spatial and temporal information efficiently and effectively. Jointly Modeling Static Visual Appearance and Temporal Pattern (JTAE) [7] jointly models static visual appearance and temporal pattern. Some advanced network architectures have been employed to mine temporal structure in videos afterward. Self-Supervised Temporal Hashing (SSTH) [37] and its extension, i.e., Self-Supervised Video Hashing (SSVH) [24] are among the pioneer unsupervised video hashing methods that model temporal sequences using LSTM. Neighborhood Preserving Hashing (NPH) [8] integrates the neighborhood attention mechanism into an RNN-based reconstruction scheme and thus enables hash code to capture spatial and temporal structure in video. Semantics-Aware Spatial-Temporal Binaries (S2Bin) [**?** ] is proposed by considering spatial-temporal context and semantic relationships, and it is applied to cross-modal video retrieval. Unsupervised Deep Video Hashing (UDVH) [32] applies a Temporal Segment Network (TSN) to extract spatial and temporal features from videos. However, training LSTM requires expensive computation and may struggle to effectively capture long-term dependencies in videos due to the gradient vanishing problem. Bidirectional Transformer Hashing (BTH) [9] utilizes a bidirectional transformer as the backbone model and designs three self-supervised learning tasks to adequately capture the similarity structure in video data. However, these hashing methods are proposed only for video-video retrieval and cannot be directly applied to more challenging video-text retrieval.

## 3 THE PROPOSED METHOD

This section first introduces the problem setup and then presents the details of the proposed Similarity Preserving Transformer Cross-Modal Hashing (SPTCH), including multi-modal semantic hash code learning and multi-modal unsupervised contrastive learning.

### 3.1 Problem Setup and Preliminary

*3.1.1 Problem Setup.* Assume that the video-text dataset has $N$ instances $O = \{O_i\}_{i=1}^N$, where $O_i = \{\mathcal{V}_i, \mathcal{T}_i\}$, $\mathcal{V}_i$ and $\mathcal{T}_i$ denote the $i$-th video and text respectively. A video is sampled with $M$ frames $\{\mathbf{f}_i^m\}_{m=1}^M$, and can be represented as CNN features $\{\mathbf{v}_i^m\}_{m=1}^M \in \mathbb{R}^{M \times d}$, where $d$ denotes CNN feature dimension. For video modality, we feed CNN features to a bidirectional transformer based hash model to obtain hash code $\mathbf{B}^v \in \{-1, 1\}^k$, where $k$ denotes

code length. For text modality, we use a pre-trained tokenizer to convert text into token sequences, and then feed to a pre-trained bidirectional transformer based hash model to obtain hash code $\mathbf{B}^t \in \{-1, 1\}^k$. The proposed SPTCH aims to learn such hash codes as compact video and text representation for video-text retrieval.

*3.1.2 Bidirectional Transformer Encoder.* Inspired by the great success of self-attention in capturing correlations in a sequence [27], we employ the bidirectional transformer to encode video and text, both of which are essentially types of sequences. For video modality, we sum the visual feature sequences $\{\mathbf{v}_i^m\}_{m=1}^M$ and position embeddings, and feed them to video transformer. Assume there are $L$ transformer layers, and each layer is constructed by multi-head attention. Specifically, in each transformer layer, given an input sequence of embedding $\mathbf{X}$, the $j$-th attention head projects $\mathbf{X}$ to a triplet of (query, key, value) denoted as $(\mathbf{Q}_j, \mathbf{K}_j, \mathbf{V}_j)$ via three learnable parameters. A scaled dot-product attention is applied between $\mathbf{Q}_j$ and $\mathbf{K}_j$, and its output is then fed to the softmax function to obtain attentional distribution over $\mathbf{V}_j$. After being passed through $L$ transformer layers, these input tokens are mapped to a sequence of l-$D$ latent visual embeddings $\{\mathbf{h}_{i,m}^v\}_{m=1}^M$. Each of such embeddings contains visual content and information flowing from other frames in both directions within the video.

For text modality, The tokenizer first splits a text corresponding to a video into words and truncates them to a unified length. Each word is assigned to a unique token in the vocabulary table, and each sentence can be transformed into a token sequence. In each sentence, we convert each token into an embedding, aggregate embeddings of all the tokens, and feed the aggregation to the bidirectional transformer. Text encoder has the same structure as video encoder.

The representation obtained through the transformer is real-valued, and a hash layer is utilized to project the continuous representation into a discrete hash code. Taking the video modality as an example, $\{\mathbf{h}_{i,m}^v\}_{m=1}^M$ is first projected as a real-valued sequence $\{\mathbf{z}_{i,m}^v\}_{m=1}^M$ via a Fully Connected (FC) layer, and then $\{\mathbf{z}_{i,m}^v\}_{m=1}^M$ is fused into a relaxed binary vector $\mathbf{z}_i^v$ by average pooling. Finally, $\mathbf{z}_i^v$ is discretized into a binary vector $\mathbf{b}_i^v$ that integrates information from all potential outputs of the transformer.

### 3.2 Multi-Modal Semantic Hash Code Learning

*3.2.1 Multi-Modal Collaborated Graph Construction.* Multiple modalities offer different properties of multi-modal data, and it is encouraging to combine multiple similarity structures from multiple modalities. For video modality, $\mathbf{X}^v$ is utilized to calculate the cosine similarity $\mathbf{S}^v$. For text modality, text features $\mathbf{h}_i^t$ are extracted using a pre-trained bidirectional transformer encoder, and we employ $\mathbf{X}^t = \{\mathbf{h}_i^t\}_{i=1}^N$ to calculate the cosine similarity $\mathbf{S}^t$. Following [39], we construct the following multi-modal collaboration graph by combining $\mathbf{S}^v$ and $\mathbf{S}^t$:

$$\mathbf{S} = \theta_1 \mathbf{S}^v + (1 - \theta_1) \mathbf{S}^t \tag{1}$$

where $\theta_1$ is used to balance the relative importance of the two modal similarity matrices. The most similar instance pairs have the top largest values in similarity matrix $\mathbf{S}$. However, the small similarity in $\mathbf{S}$ are more likely to be affected by noise and may hinder unsupervised learning. To eliminate such noisy effects, we

set the minimum $\theta_2\%$ in each row of $\mathbf{S}$ to -1 with the following equation:

$$S_{ij} = \begin{cases} -1, S_{ij} \epsilon e_i \, (\theta_2) \\ S_{ij}, \text{otherwise} \end{cases} \tag{2}$$

where $e_i \, (\theta_2)$ is the set consisting of the minimum $\theta_2\%$ values in the $i$-th row of $\mathbf{S}$. We obtain the final multi-modal collaboration graph by applying tanh function on $\mathbf{S}$ for nonlinear transformation.

*3.2.2 Cross-Modal Semantic Aggregation.* The video and text features $\mathbf{X}^v$ and $\mathbf{X}^t$ are fed into two encoders to obtain latent semantic representations $\mathbf{V}^v$ and $\mathbf{V}^t$ respectively. In this work, inspired by the superior capability of GCN on capturing higher-order semantic similarity structure, we employ two modality-specific GCNs on $\mathbf{V}^v$ and $\mathbf{V}^t$ to obtain graph structure representations $\mathbf{V}^{gv}$ and $\mathbf{V}^{gt}$ respectively. The layer-wise propagation rule of modality-specific GCNs is defined as:

$$\mathbf{H}_{(l)} = \sigma \left( \widetilde{\mathbf{D}}^{-\frac{1}{2}} \mathbf{S} \widetilde{\mathbf{D}}^{-\frac{1}{2}} \mathbf{H}_{(l-1)} \mathbf{W}_{(l)} \right) \tag{3}$$

where $\widetilde{\mathbf{D}}$ is a diagonal matrix, $\widetilde{\mathbf{D}}_{ii} = \sum_i S_{ij}$. The latent video and text features $\mathbf{V}^v$ and $\mathbf{V}^t$ are concatenated into a new representation $\mathbf{V}^c$, which is then fed into a multi-modal GCN to enhance the interactions between heterogeneous modal features. The layer-wise propagation rule of such multi-modal GCN is defined as:

$$\mathbf{H}^c_{(l)} = \sigma \left( \widetilde{\mathbf{D}^c}^{-\frac{1}{2}} \mathbf{S}^c \widetilde{\mathbf{D}^c}^{-\frac{1}{2}} \mathbf{H}^c_{(l-1)} \mathbf{W}^c_{(l)} \right) \tag{4}$$

$$\mathbf{S}^c = \begin{pmatrix} \mathbf{S} & \mathbf{I} \\ \mathbf{I} & \mathbf{S} \end{pmatrix}, \mathbf{H}^c_{(l-1)} = \begin{pmatrix} \mathbf{V}^v_{(l-1)} \\ \mathbf{V}^t_{(l-1)} \end{pmatrix} \tag{5}$$

where $\widetilde{\mathbf{D}^c}$ is a diagonal matrix, $\widetilde{\mathbf{D}^c}_{ii} = \sum_i S^c_{ij}$, $\mathbf{H}^c_{(0)} = \mathbf{V}^c$. Multi-modal GCN performs semantic aggregation between fused features, preserves multi-modal neighborhood correlations, and thus reduces the modality gap.

*3.2.3 Loss.* We propose to employ a fully connected layer to aggregate $\mathbf{V}^j$ and $\mathbf{V}^{gv}$ to generate the latent representation $\mathbf{H}^v$ that contains more semantics among different modalities. We then fed $\mathbf{H}^v$ into the video decoder to reconstruct the original video feature $\mathbf{X}^v$. In the text branch, we generate $\mathbf{H}^t$ similar to $\mathbf{H}^v$, and further reconstruct $\mathbf{X}^t$. The semantic reconstruction loss of the video and text modalities is defined as:

$$\mathcal{L}_{rc} = \|\widetilde{\mathbf{X}^v} - \mathbf{X}^v\|_F^2 + \|\widetilde{\mathbf{X}^t} - \mathbf{X}^t\|_F^2 \tag{6}$$

where $\widetilde{\mathbf{X}^v}$ and $\widetilde{\mathbf{X}^t}$ denote the reconstructed video and text features, respectively.

In this work, $\mathbf{H}^v$, $\mathbf{H}^t$, and $\mathbf{V}^j$ contain different semantics and are complementary. We first employ a fully connected layer to project $\mathbf{V}^j$ to obtain $\mathbf{H}^j$, and concatenate $\mathbf{H}^j$, $\mathbf{H}^v$, and $\mathbf{H}^t$ to obtain a composite multi-modal complementary representation $\mathbf{H}$. SPTCH expects that the generated latent semantic representations $\mathbf{H}^v$, $\mathbf{H}^t$ and $\mathbf{H}$ can reflect the correlations among multi-modal data and can preserve similarity structure well. Specifically, we employ cosine similarity among the latent representations to approximate the ground-truth similarity. To achieve this, we minimize the following neighborhood reconstruction loss:

$$\mathcal{L}_{sc} = \|\mathbf{S} - \cos(\mathbf{H}^v, \mathbf{H}^v)\|_F^2 + \|\mathbf{S} - \cos(\mathbf{H}^t, \mathbf{H}^t)\|_F^2 + \|\mathbf{S} - \cos(\mathbf{H}, \mathbf{H})\|_F^2 \tag{7}$$

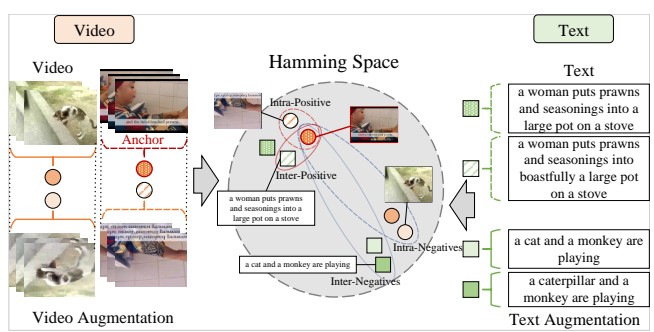

**Figure 2: Illustration of multi-modal contrastive learning. We construct inter- and intra-modal triplet sets. Based on the two sets, we define inter- and intra-modal contrastive losses to preserve inter- and intra-modal similarity structures respectively.**

We minimize the following quantization loss to enable continuous network outputs and hash codes to be close:

$$\mathcal{L}_{bc} = \|\mathbf{B} - \mathbf{H}\|_F^2 \tag{8}$$

To this end, by summarizing the above three losses, i.e., $\mathcal{L}_{rc}$, $\mathcal{L}_{sc}$, and $\mathcal{L}_{bc}$, we have the following objective function of multi-modal semantic hashing learning:

$$\mathcal{L}_c = \mathcal{L}_{rc} + \alpha_1 \mathcal{L}_{sc} + \alpha_2 \mathcal{L}_{bc} \tag{9}$$

where the two parameters $\alpha_1$ and $\alpha_2$ are used to balance each term.

## 3.3 Multi-Modal Unsupervised Contrastive Learning

The learned multi-modal semantic hash code contains neighborhood information of the original space. This section introduces multi-modal unsupervised contrastive learning to effectively exploit the inter-modal and intra-modal similarity structure among videos and texts. The general goal of contrastive learning is to pull the anchor and positive sample together and push apart the anchor from negative samples. The overview of the proposed multi-modal unsupervised contrastive learning is illustrated in Figure 2.

Given a set of $N$ samples $\{\mathbf{x}_i, \mathbf{g}_i\}_{i=1}^N$, $\mathbf{g}_i$ is the label vector of $\mathbf{x}_i$. Its augmented set is denoted as $\{\hat{\mathbf{x}}_i, \mathbf{g}_i\}_{i=1}^{2N}$ that has $2N$ samples, and $\hat{\mathbf{x}}_{2i-1}$ and $\hat{\mathbf{x}}_{2i}$ are two random augmentations of $\mathbf{x}_i (i = 1 \dots N)$. Assume that $i \in I = \{1 \dots 2N\}$ denotes the index of an arbitrary augmented sample, and $(\hat{\mathbf{x}}_i, \hat{\mathbf{x}}_j)$ denotes a positive pair. The conventional self-supervised contrastive loss [2] is defined as follows:

$$\mathcal{L}_{self} = -\frac{1}{2N} \sum_{i=1}^{2N} \log \frac{\exp(\hat{\mathbf{x}}_i \cdot \hat{\mathbf{x}}_j / \tau)}{\sum_{a \in \mathbf{A}(i)} \exp(\hat{\mathbf{x}}_i \cdot \hat{\mathbf{x}}_a / \tau)} \tag{10}$$

where $\tau$ is a temperature coefficient that controls the dynamic range of the product, $\mathbf{A}(i) = \{a | a \in I, a \neq i\}$, $\hat{\mathbf{x}}_i$ and $\hat{\mathbf{x}}_j$ are called the anchor and positive respectively, and the other $2(N - 1)$ samples $\{\hat{\mathbf{x}}_k | \hat{\mathbf{x}}_k \in I, k \neq i, k \neq j\}$ are called the negatives.

*3.3.1 Inter-Modal Contrastive Loss.* Inter-modal contrastive learning is defined based on a triplet set of anchor, inter-positive, and

**Algorithm 1** Similarity Preserving Transformer Cross-Modal Hashing

**Input**: video-text pairs $O_i = \{\mathcal{V}_i, \mathcal{T}_i\}_{i=1}^N$; input dimension $d$; code length $k$; batch size; number of epochs; learning rate; parameters.
**Output**: network parameters.
1: Extract video and text features $\mathbf{X}^v$ and $\mathbf{X}^t$;
2: Construct multi-modal collaboration graph $\mathbf{S}$;
3: **for** each epoch **do**
4:     **for** each iteration **do**
5:         Sample a minibatch randomly;
6:         Obtain $\mathbf{H}^v$, $\mathbf{H}^t$, and $\mathbf{H}^j$ via forward propagation algorithm;
7:         Calculate loss via (6), (7), and (8);
8:         Update multi-modal semantic hashing network parameters to minimize (9) via BP algorithm;
9:     **end for**
10:     **for** each iteration **do**
11:         Sample a minibatch randomly;
12:         Obtain $\mathbf{B}$ via GCN and encoder;
13:         Obtain $\{\mathbf{h}_{i,m}^v\}_{m=1}^M$ and $\mathbf{h}_i^t$ via bidirectional transformer;
14:         Calculate multi-modal unsupervised contrastive loss $\mathcal{L}_{mf}$ via (13);
15:         Update bidirectional transformer network parameters to minimize (16) via BP algorithm;
16:     **end for**
17: **end for**

inter-negative samples. Inter-modal contrastive learning simultaneously encourages the embedding of anchors to be close to that of inter-positive samples and to be far away from those of inter-negative samples, such that cross-modal correlation can be effectively exploited in latent embedding space.

Let $\mathbf{z}_i^v$ and $\hat{\mathbf{z}}_i^v$ be original and augmented features of the $i$-th video. Similarly, $\mathbf{z}_i^t$ and $\hat{\mathbf{z}}_i^t$ are the original and augmented features of the $i$-th text. By regarding $\mathbf{z}_i^v$ and $\hat{\mathbf{z}}_i^v$ as anchors, we have the following inter-modal contrastive loss:

$$
\begin{aligned}
\mathcal{L}_m^{ve} = &\frac{1}{N}\sum_{i=1}^N -\log \frac{\exp\left(\mathbf{z}_i^v \cdot \mathbf{z}_i^t/\tau\right)}{\sum_{a\in\mathbf{A}(i)}\exp\left(\mathbf{z}_i^v \cdot \mathbf{z}_a^t/\tau\right)} \\
&+ \frac{1}{N}\sum_{i=1}^N -\log \frac{\exp\left(\hat{\mathbf{z}}_i^v \cdot \hat{\mathbf{z}}_i^t/\tau\right)}{\sum_{a\in\mathbf{A}(i)}\exp\left(\hat{\mathbf{z}}_i^v \cdot \hat{\mathbf{z}}_a^t/\tau\right)}
\end{aligned}
\tag{11}
$$

Accordingly, inter-modal contrastive loss $\mathcal{L}_m^{te}$ for text modality can be similarly defined.

*3.3.2 Intra-Modal Contrastive Loss.* Intra-modal contrastive learning defines a triplet set of anchor, intra-positive, and intra-negative samples. Intra-modal contrastive learning enables the embedding of anchors to be close to that of intra-positive samples and far away from those of intra-negative samples, thereby preserving the intrinsic similarity structure within each modality. Taking the

video modality as an example, we have the following intra-modal contrastive loss:

$$
\mathcal{L}_m^{va} = \frac{1}{N}\sum_{i=1}^N -\log \frac{\exp\left(\mathbf{z}_i^v \cdot \hat{\mathbf{z}}_i^v/\tau\right)}{\sum_{a\in\mathbf{A}(i)}\exp\left(\mathbf{z}_i^v \cdot \hat{\mathbf{z}}_a^v/\tau\right)}
\tag{12}
$$

Accordingly, intra-modal contrastive loss $\mathcal{L}_m^{ta}$ for text modality can be similarly defined. To sum up, we have the following multi-modal unsupervised contrastive learning loss:

$$
\mathcal{L}_{mf} = \mathcal{L}_m^{ve} + \mathcal{L}_m^{te} + \mathcal{L}_m^{va} + \mathcal{L}_m^{ta}
\tag{13}
$$

In addition, we enforce the outputs of bidirectional transformer encoders to be close to the learned hash code. We minimize the following loss:

$$
\mathcal{L}_{bf} = \|\mathbf{B} - \mathbf{Z}^v\|_F^2 + \|\mathbf{B} - \mathbf{Z}^t\|_F^2
\tag{14}
$$

where $\mathbf{Z}^v = \left\{\mathbf{z}_i^v\right\}_{i=1}^N$ and $\mathbf{Z}^t = \left\{\mathbf{z}_i^t\right\}_{i=1}^N$ represent video and text features respectively.

*3.3.3 Neighborhood Preservation Loss.* In addition to preserving similarity introduced by contrastive learning, we further expect to use cosine similarity of the outputs of the bidirectional transformer encoder to approximate the ground-truth similarity. To achieve this, we minimize the following neighborhood preservation loss:

$$
\begin{aligned}
\mathcal{L}_{sf} = &\|\mathbf{S} - \cos\left(\mathbf{Z}^v, \mathbf{Z}^t\right)\|_F^2 + \|\mathbf{S} - \cos\left(\mathbf{Z}^v, \mathbf{Z}^v\right)\|_F^2 \\
&+ \|\mathbf{S} - \cos\left(\mathbf{Z}^t, \mathbf{Z}^t\right)\|_F^2
\end{aligned}
\tag{15}
$$

To this end, by summarizing the above three losses, i.e., $\mathcal{L}_{mf}$, $\mathcal{L}_{bf}$, and $\mathcal{L}_{sf}$, we have the following objective function of SPTCH hashing encoder:

$$
\mathcal{L}_f = \mathcal{L}_{mf} + \beta_1 \mathcal{L}_{bf} + \beta_2 \mathcal{L}_{sf}
\tag{16}
$$

where $\beta_1$ and $\beta_2$ are the two parameters to balance each term. SPTCH optimizes the entire network using backpropagation. The training procedure of the proposed SPTCH is illustrated in Algorithm 1.

## 4 EXPERIMENTS

This section evaluates the performance of the proposed method for video-text retrieval. The experiments are performed on an Ubuntu Enterprise 64-bit Linux workstation with 128G memory and a NVIDIA A6000 GPU server.

### 4.1 Experimental Setup

*4.1.1 Datasets.* The experiments are conducted on three benchmark video text datasets, which have been widely used for video-text analysis. The three datasets are detailed as follows:

MSR-VTT [33] is the largest general video captioning dataset. It contains 10,000 video clips with 41.2 hours and 200,000 clip-sentence pairs in 20 categories. Additionally, each video clip has been manually annotated with 20 natural sentences. Following [37], we randomly choose 6,513 and 2,990 clips for training and testing respectively.

ActivityNet Captions v1.2 [6] is a large-scale video dataset for human action understanding. It contains more than 13,000 videos

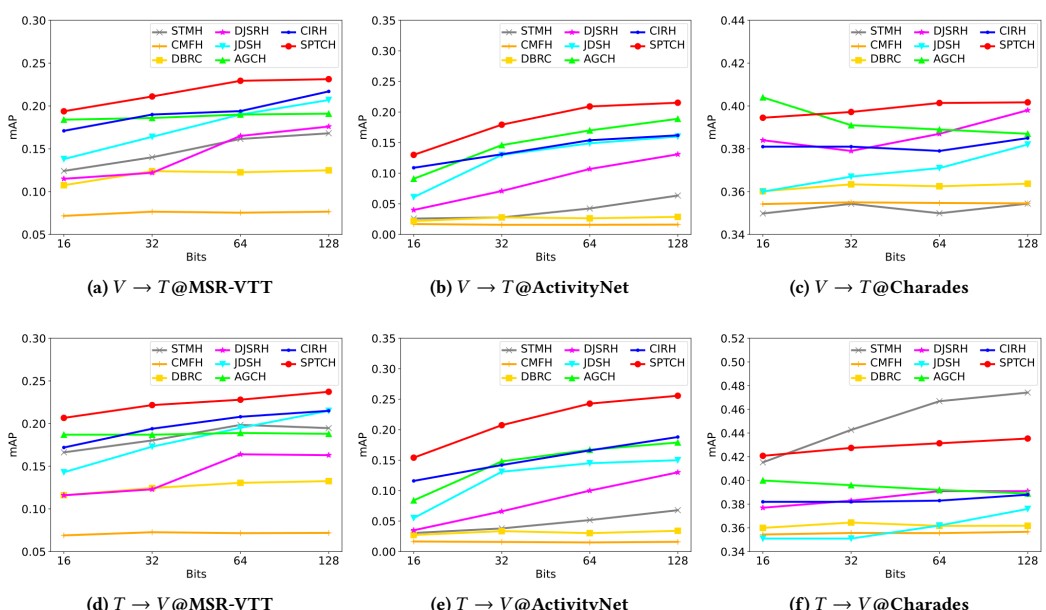

Figure 3: mAPs of all the hashing methods with respect to different code lengths in two video-text retrieval tasks.

from 100 activity categories collected from YouTube, with an average of 137 untrimmed videos per class and 1.41 activity instances per video. We randomly choose 4,816 and 2,382 videos for training and testing respectively.

Charades [21] is a dataset composed of 9848 videos of daily indoor activities collected through Amazon Mechanical Turk. The dataset contains 66,500 temporal annotations for 157 action classes, 41,104 labels for 46 object classes, and 27,847 textual descriptions of the videos. Since the test set does not provide labels, we use the validation set for testing. We choose 7,985 and 1,863 videos for training and testing respectively.

*4.1.2    Baselines.* To our knowledge, there are few cross-modal hashing methods specifically designed for video-text retrieval. We compare SPTCH with seven state-of-the-art hashing methods, including two shallow cross-modal hashing methods, i.e., CMFH [3], STMH [28], five deep cross-modal hashing methods, i.e., CIRH [39], AGCH [38], JDSH [16], DJSRH [25], and DBRC [11]. For CMFH and STMH, following [22], we apply mean pooling on features extracted by VGG-16 to represent video. For CIRH, AGCH, JDSH, DJSRH, and DBRC, we employ I3D [1] as the backbone to encode video.

*4.1.3    Experiment Setting.* Following [37], we first sample 25 frames resized to 224×224 for each video and extract 4096-$D$ frame features with VGG-16 [22] pre-trained on ImageNet [20]. The video transformer includes four layers with 256-$D$ attention head, and the scaling factor $d_k$ is set to 256. We first concatenate all texts belonging to the same video and then tokenize concatenated text as input. The text transformer has the same structure as the video transformer, and its pre-trained model is provided by Hugging Face. The batch size, number of epochs, GCN learning rate, and transformer learning rate are set to 128, 20, $1 \times 10^{-3}$, and $1 \times 10^{-4}$

respectively. The parameters $\alpha_1$, $\alpha_2$, $\beta_1$, and $\beta_2$ are empirically set to 0.2, 0.2, 1, and 1 respectively. In the construction of multi-modal graphs, the parameters $\theta_1$ and $\theta_2$ are empirically set to 0.6 and 0.1 respectively. The temperature coefficient is empirically set to 0.2. The proposed SPTCH is optimized using Adam optimizer.

*4.1.4    Data Augmentation.* In this work, we apply the same spatial augmentation to frames for video augmentation to avoid temporal structure disruption. We collect the augmented frames that do not overlap with the original frames to capture a wider range of information. Specifically, we apply random cropping, flipping, noise, color-jittering, and blurring on each collected frame, where the parameters are first randomly generated for each video and then applied to all the frames.

We employ the widely-used Easy Data Augmentation (EDA) [29] for text augmentation. Given a sentence, we apply four augmentation approaches randomly, including (1) Synonym Replacement(SR) that randomly replaces n non-stop-words with synonyms, (2) Random Insertion (RI) that randomly inserts a synonym of a non-stop-words into a random position, (3) Random Swap (RS) that randomly swaps two words, (4) Random Deletion (RD) that randomly removes each word with a probability.

*4.1.5    Evaluation Metric.* Following [37], we consider the widely-used mean Average Precision (mAP) and Precision-Recall (PR) curve as evaluation metrics [18].

## 4.2    Comparsions With State-of-The-Arts

This section evaluates the proposed SPTCH by comparing it with state-of-the-art cross-modal hashing methods in video-text retrieval. Figure 3 reports the mAPs of all the hashing methods in two video-text retrieval tasks on three benchmark datasets. In addition, the

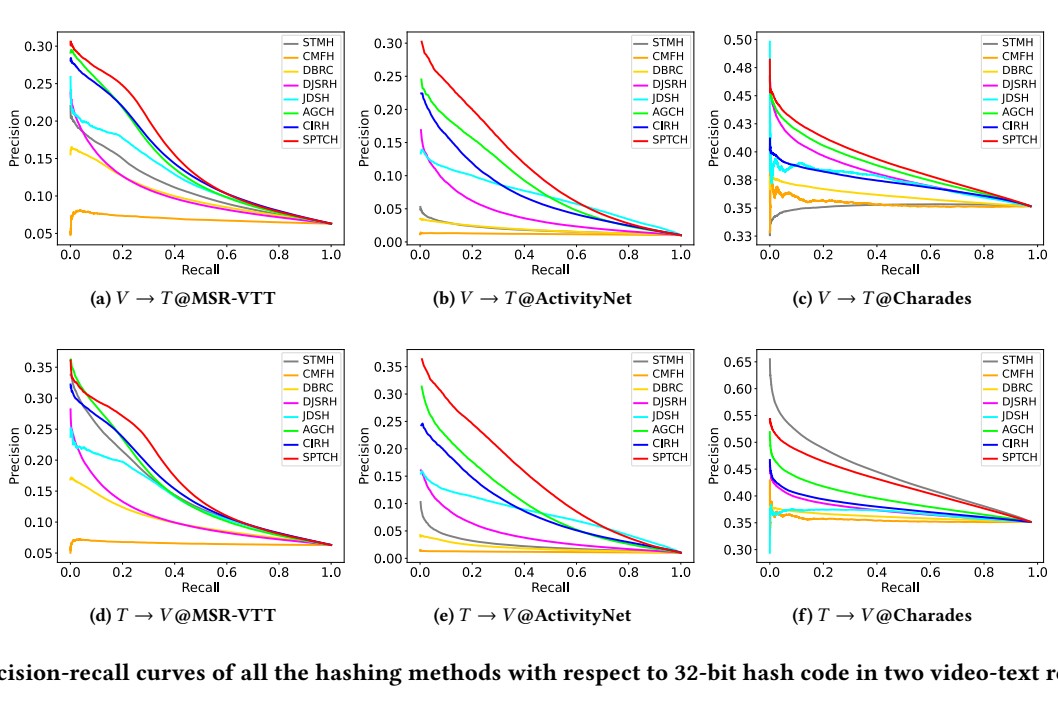

Figure 4: Precision-recall curves of all the hashing methods with respect to 32-bit hash code in two video-text retrieval tasks.

Precision-Recall (PR) curves with respect to 32 bits are shown in Figure 4. From the above results, we can clearly observe that (1) the proposed SPTCH outperforms all the baselines in most cases. For instance, for MSR-VTT, SPTCH improves the best baseline by 2.11%, 3.54%, and 1.43% with 32, 64, and 128 bits respectively in $V \rightarrow T$ task, and improves the best baseline by 2.77%, 2%, and 2.23% with 32, 64, and 128 bits respectively in $T \rightarrow V$ task. The PR curves of the proposed SPTCH are generally above those of the most baselines, demonstrating again the effectiveness of the proposed method on video-text retrieval tasks. (2) Among all the baselines, deep hashing methods generally outperform shallow hashing methods in most cases. (3) Among deep hashing baselines, CIRH performs best, indicating its superior capabilities of capturing multi-modal semantic structure.

## 4.3 Further Analysis

*4.3.1 Ablation Study.* This section first conducts an ablation study of the proposed method by analyzing the effectiveness of each of the three losses. Specifically, we compare the proposed method with the following three variants, including (1) SPTCH-Cont: a variant that removes multi-modal unsupervised contrastive loss; (2) SPTCH-Sim: a variant that removes similarity retention loss; (3) SPTCH-Bin: a variant that removes multi-modal semantic hash code learning loss. We adopt MSR-VTT for the experiment and report the mAPs of these methods in Table 1. As can be observed from Table 1, SPTCH significantly outperforms SPTCH-Cont, demonstrating the importance of multi-modal unsupervised contrastive loss. In addition, SPTCH further improves SPTCH-Sim and SPTCH-Bin, verifying the effectiveness of similarity retention loss and multi-modal semantic hash code learning loss.

Table 1: Ablation study of the proposed SPTCH on MSR-VTT.

| Method | $V \rightarrow T$ | | $T \rightarrow V$ | |
|---|---|---|---|---|
| | 32 bits | 64 bits | 32 bits | 64 bits |
| SPTCH-Cont | 0.1444 | 0.1559 | 0.1395 | 0.1563 |
| SPTCH-Sim | 0.1984 | 0.206 | 0.1968 | 0.2042 |
| SPTCH-Bin | 0.1993 | 0.2167 | 0.1864 | 0.2004 |
| SPTCH-S1 | 0.1923 | 0.1967 | 0.1957 | 0.2039 |
| SPTCH-B1 | 0.1975 | 0.2127 | 0.2039 | 0.2090 |
| SPTCH-B2 | 0.1942 | 0.2107 | 0.2002 | 0.2142 |
| SPTCH | **0.2111** | **0.2294** | **0.2217** | **0.2280** |

This section then conducts an ablation study by analyzing the effectiveness of some operations on the multi-modal collaborative graph. Specifically, we compare the proposed method with the following three variants, including (1) SPTCH-S1: a variant that removes noise processing and normalization operation on S; (2) SPTCH-B1: a variant that removes cross-modal semantic aggregation operation such that $\mathbf{H}$ is generated by concatenating $\mathbf{H}^v$ and $\mathbf{H}^t$; (3) SPTCH-B2: a variant that removes all the GCN modules. From Table 1, we can observe that the proposed SPTCH outperforms the three variants. It reveals that graph processing, graph aggregation, and GCN lead to improved similarity structure preservation, providing retrieval performance.

*4.3.2 Parameter Analysis.* The section analyzes the sensitivity of four important trade-off parameters in the proposed method, i.e., $\alpha_1$, $\alpha_2$, $\beta_1$, and $\beta_2$, where $\alpha_1$, $\alpha_2$ are varied from [0.01, 1], and $\beta_1$, $\beta_2$ are varied from [0.01, 10]. We adopt MSR-VTT for the experiment, set the code length to 64, and report the mAPs with respect to different parameters in Figure 6. As can be observed, as the four parameters increase, the mAPs first increase stably and then decrease. The

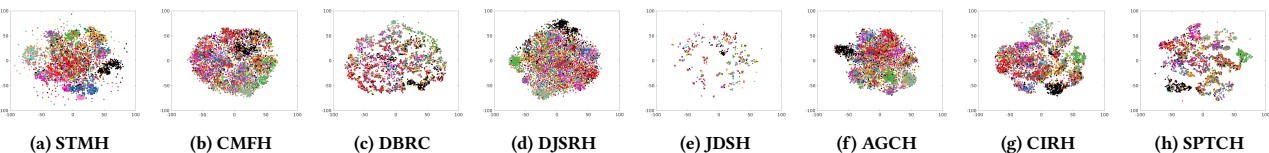

**Figure 5: The t-SNE visualization of MSR-VTT using all the hashing methods.**

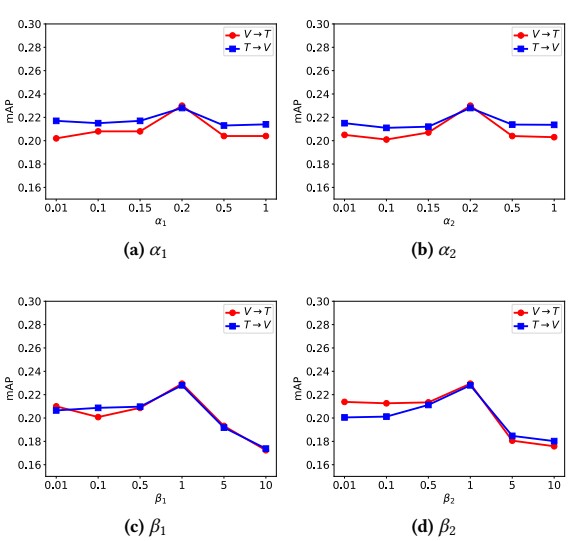

**(a) $\alpha_1$**

**(b) $\alpha_2$**

**(c) $\beta_1$**

**(d) $\beta_2$**

**Figure 6: Parameter analysis of the proposed SPTCH on MSR-VTT.**

highest mAPs of the proposed SPTCH are obtained when $\alpha_1$ and $\alpha_2$ are set to 0.2 and $\beta_1$ and $\beta_2$ are set to 1.

*4.3.3 Visualization.* This section visualizes learned hash code to qualitatively compare different hashing methods. The MSR-VTT is adopted for the experiment, the code length is set to 32, and only the samples annotated with a single label are selected. The hash codes learned by all the hashing methods are visualized into a 2-dimensional space with t-SNE [26], as illustrated in Figure 5. From Figure 5, we see that visualization is generally consistent with quantitative empirical results, and SPTCH provides better visualization results than the baselines.

*4.3.4 Case Study.* This section presents a case study on video-text retrieval, the proposed method, and the competitive baseline, i.e., CIRH experiment on MSR-VTT. Figure 7 illustrates the top-5 retrieved results of one randomly selected query video and text. The correct and incorrect retrieved results are marked by ticks and crosses respectively. From this figure, we see that the proposed SPTCH is capable of retrieving more correct results than CIRH.

## 5 CONCLUSION

In this work, we propose a new unsupervised deep cross-modal hashing method designed for video-text retrieval. SPTCH exploits

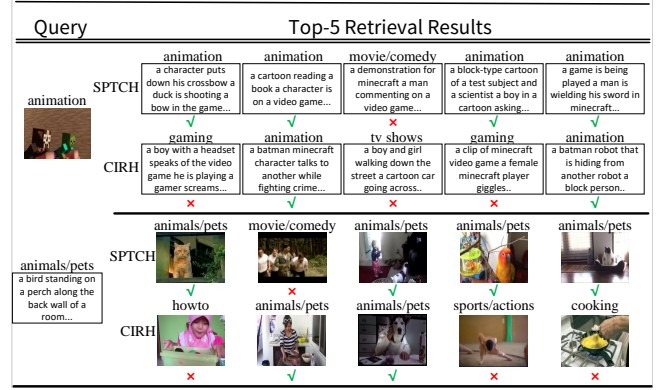

**Figure 7: Top-5 retrieved results of the proposed SPTCH and CIRH on one randomly selected query video and text pair from MSR-VTT.**

long-term dependencies by encoding video and text via bidirectional Transformer. SPTCH applies semantic aggregation by employing GCN on constructed multi-modal collaborative graph. SPTCH exploits inter- and intra-modal similarity structure using unsupervised multi-modal contrastive loss and neighborhood reconstruction loss. Extensive empirical results demonstrate superiority of the proposed method and verify effectiveness of each component. In the future, we aim to develop cross-modal hashing for degraded video-text retrieval where video and text are of inferior quality.

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
