# OpenReview forum: "Similarity Preserving Transformer Cross-Modal Hashing for Video-Text Retrieval"
_acmmm.org/ACMMM/2024/Conference — MM2024 Poster_

### Official Review · Reviewer_1SZS · 2024-05-20

**Rating:** 5
**Confidence:** 3

**Summary:**

This paper studies video-text retrieval that is more challenging than image-text retrieval. This work proposes Similarity Preserving Transformer Cross-Modal Hashing (SPTCH) to address the video-text retrieval task, which is challenging yet less studied in hashing. To explore the long-term dependencies in videos and texts, the proposed method employs Vision Transformer on the two modalities. To exploit structure among videos and texts, the proposed method constructs a multi-modal collaborative graph, and employs GCN and contrastive loss to preserve similarity in an unsupervised manner.

**Strengths:**

1) This paper is well written, and is very easy to follow.
2) In the areas of unsupervised deep cross-modal hashing, the focused problem video-text retrieval is more challenging than widely-studied image-text retrieval, as video is more complex than image. The focused study is meaningful.
3) The used techniques are novel in hashing and sound.
4) They provide empirical comparisons with state-of-the-arts, and show its superiority of the proposed method.

**Limitations:**

1) This paper does not show its limitation of this work.
2) One reference in Line 243 is cited properly. Should correct it.
3) I wonder that the importance of contrastive loss contributing to the performance.

**Suitability:**

3

---

### Official Review · Reviewer_B8nv · 2024-05-24

**Rating:** 2
**Confidence:** 3

**Summary:**

This paper proposes an unsupervised multi-modal contrastive loss and a neighborhood reconstruction loss to effectively exploit inter and intra modal similarity structure among videos and texts. Several experiments are conducted to show the effectiveness of the proposed method.

**Strengths:**

1.  The paper is written and organized well, which is easy to follow and understand even for the beginners.
2. The ablation experiments and hyperparameter experiments are conducted to show the effectiveness of each component of the method.

**Limitations:**

1. Figure 1 is not neat enough, even the alignments among blocks are not achieve. The data flow too complex and not clear. It also exists some missing citing references.
3. In section 3.1.2, it is not necessary to use such a long page to introduce the background knowledge that is common and widely known.
4. Common cross-modal hashing methods usually have experiments on different length of hash code, but the authors do not have the experiments on different hash code length.
5. The paper lacks of novelty and workload, the methods in section 3.2 and 3.3 are very common in multi-modal area. For example, ‘Specific class center guided deep hashing for cross-modal retrieval’. It is just some easy stacking of existing methods.
6. The compared methods are slightly outdated, only CIRH method is a relatively new method. More comparative experiments should be conducted with recently proposed methods.

**Suitability:**

2

---

### Official Review · Reviewer_KvKA · 2024-05-25

**Rating:** 4
**Confidence:** 1

**Summary:**

This paper proposes a Similarity Preserving Transformer Cross-Modal Hashing (SPTCH) method for video-text retrieval. SPTCH encodes video and text using bidirectional transformers, constructs a multi-modal collaborative graph to aggregate semantics, and learns hash codes by minimizing semantic reconstruction loss, neighborhood reconstruction loss, and multi-modal contrastive loss

**Strengths:**

1. SPTCH effectively captures long-term dependencies in video and text modalities by employing bidirectional transformer encoders, which is crucial for understanding the semantic content in both modalities.

2. Extensive experiments on three benchmark datasets demonstrate the superiority of SPTCH over state-of-the-art methods in video-text retrieval tasks.

**Limitations:**

1. The ablation study could be more comprehensive to better understand the individual contributions of the key components in SPTCH, such as the bidirectional transformer encoders and the multi-modal collaborative graph.
2. The paper does not provide a qualitative analysis of the learned hash codes to gain insights into how SPTCH captures the semantic relationships between videos and texts.

**Suitability:**

3

---

### Meta-Review · Area_Chair_gthn · 2024-07-02

**Recommendation:** Accept (Poster)
**Confidence:** 4

**Metareview:**

Cross-modal video-text retrieval is still in need of good solutions. This paper has a reasonable idea and approach. he empirical results are strong (at least for the chose benchmark datasets).

I am in agreement with the author(s) in their response/comment on the reviewer who gave a lower rating: some of his/her negative comments are not well supported as the paper already addressed the issues raised.